# Spatially and cell-type resolved quantitative proteomic atlas of healthy human skin

Beatrice Dyring-Andersen[1,2,3,4], Marianne Bengtson Løvendorf[5], Fabian Coscia[1], Alberto Santos[1], Line Bruun Pilgaard Møller[1], Ana R. Colaço[1], Lili Niu[1], Michael Bzorek[6], Sophia Doll[7], Jørgen Lock Andersen[8], Rachael A. Clark[2], Lone Skov [4], Marcel B. M. Teunissen [9] & Matthias Mann [1,7✉]

Human skin provides both physical integrity and immunological protection from the external environment using functionally distinct layers, cell types and extracellular matrix. Despite its central role in human health and disease, the constituent proteins of skin have not been systematically characterized. Here, we combine advanced tissue dissection methods, flow cytometry and state-of-the-art proteomics to describe a spatially-resolved quantitative proteomic atlas of human skin. We quantify 10,701 proteins as a function of their spatial location and cellular origin. The resulting protein atlas and our initial data analyses demonstrate the value of proteomics for understanding cell-type diversity within the skin. We describe the quantitative distribution of structural proteins, known and previously undescribed proteins specific to cellular subsets and those with specialized immunological functions such as cytokines and chemokines. We anticipate that this proteomic atlas of human skin will become an essential community resource for basic and translational research (https://skin.science/).

[1] Novo Nordisk Foundation (NNF) Center for Protein Research, Faculty of Health and Medical Sciences, University of Copenhagen, Copenhagen, Denmark. [2] Department of Dermatology, Brigham and Women's Hospital and Harvard Medical School, Boston, USA. [3] Leo Foundation Skin Immunology Research Center, Faculty of Health and Medical Sciences, University of Copenhagen, Copenhagen, Denmark. [4] Department of Dermatology and Allergology, Herlev and Gentofte Hospital, University of Copenhagen, Hellerup, Denmark. [5] Center for RNA Medicine, Department of Clinical Medicine, Aalborg University, Copenhagen, Denmark. [6] Department of Surgical Pathology, Zealand University Hospital, Næstved, Denmark. [7] Department of Proteomics and Signal Transduction, Max Planck Institute of Biochemistry, Martinsried, Germany. [8] Department of Plastic and Breast Surgery, Zealand University Hospital, Roskilde, Denmark. [9] Department of Dermatology, Amsterdam University Medical Centers, location AMC, Amsterdam, Netherlands. ✉email: mmann@biochem.mpg.de

Human skin serves as the interface between the body and the surrounding environment, and, as such, it provides essential functions such as sensing and immunological protection. The skin also provides physical protection against injury and assists in the control of body temperature and our perception of the surrounding environment through sensory nerves. Accordingly, the skin is a complex organ comprising multiple tissue layers and diverse cell types, including melanocytes, endothelial cells, keratinocytes, fibroblasts, and skin-associated immune cells. It also contains a high density of extracellular matrix (ECM) that contributes to its tensile strength and flexibility. More than a century of research on the skin and its structure has provided an increasingly granular view of this complex organ, including its micro-anatomy and the multitude of immune and other cells it contains[1–5]. Recently, transcriptomic profiling studies of skin, including analyses at the single-cell level, have mapped the gene expression landscape of skin-associated cells and revealed the presence of skin-specific genes[6–8]. This suggests that the skin and its component cell types likely employ unique proteomes to facilitate the unique structural, homeostatic, and immune functions of skin.

Despite its importance, there have been no prior comprehensive and quantitative studies of the molecular composition of healthy human skin at the protein level. Previous studies have identified a small fraction of the expressed proteome, likely because of difficulties in separating and homogenizing tissue and the dominance of structural proteins such as keratin (KRT)[9–11]. Recent technological advances in the field of mass spectrometry (MS)-based proteomics have dramatically improved the sensitivity, throughput, completeness and quantitative accuracy of this technique[12–15]. MS-based proteomics have now become a sensitive and accurate method for large-scale, unbiased proteomic analyses that enable characterization of nearly complete proteomes[16–18].

Here, we combine MS-based proteomics with an advanced sampling strategy that accounts for the cellular complexity of the skin. This approach yields a highly resolved quantitative proteomic atlas of the different layers and cell types in healthy human skin, allowing us to measure and map the quantitative distribution of functional and structural proteins. As a first example of how this atlas can be used, we analyze immune cell types and proteins with known immunologic functions, yielding a spatial and cellular map of cytokine production. This proteomic atlas provides a comprehensive characterization of the quantitative distribution and relationships of proteins in human skin. It therefore represents a valuable resource for future systematic studies of the biology and pathology of human skin.

## Results

**An in-depth quantitative proteomic atlas of human skin.** To create a layer and cell-type resolved proteomic atlas of the human skin, we used healthy skin from plastic surgeries (see "Methods"). Initially, we removed stratum corneum with tape stripping; however, we found that this method did not allow us to sample the inner epidermis. Instead, we dissected inner and outer epidermis by curettage and subsequently used punch biopsies for dermis and subcutis (Fig. 1a). This procedure cleanly divided the layers as judged by histology (Supplementary Fig. 1a–c). In addition, we established primary cell cultures of keratinocytes and fibroblasts from the same skin donors (see "Methods")[19]. Finally, we isolated immune cells, melanocytes and endothelial cells by enzymatic digestion followed by FACS (Supplementary Fig. 2). Together, these samples comprise the complete depth of the skin tissue and its major cell types. We derived the tissue material from six individuals, whereas immune cells were obtained from

up to 21 donors, resulting in a total of more than 60 proteome samples (see Methods). From each of them, we extracted proteins through a combination of tissue homogenization, heating and ultrasonication, followed by overnight tryptic digestion.

To construct an inventory of skin proteins that's as comprehensive as possible, we pooled all samples across donors and separated them into eight or 16 fractions by high-pH reversed-phase fractionation[20]. All samples were measured on a quadrupole orbitrap mass spectrometer in 100 min data-dependent acquisition mode (DDA) (Fig. 1a–b, Q Exactive HF-X, Methods). The resulting proteome library comprised 173,228 sequence-unique peptides accounting for 10,701 proteins in total with a median protein sequence coverage of 39.4% (Supplementary Data 1). The proteins that were quantified in the present study spanned more than six orders of magnitude. Given the challenging nature of this tissue, this is an unprecedented depth of coverage, providing an excellent basis to study skin function and composition on a global proteomic level.

This proteome library served as a deep spectral library for all data-independent acquisition (DIA) runs, which we performed separately on the three to six donor replicates for each skin layer and cell type, resulting in a total of 60 DIA proteome measurements (Fig. 1c). This was particularly valuable for the immune cell types, where sample amounts were very limited. In the fractionated proteomes, we identified 8987 proteins in the outer epidermis (stratum corneum) and 9140 proteins in the inner epidermis (comprising cell layers down to basal membrane). In the dermis and in the subcutis, we identified a somewhat lower number of proteins (8340 and 6334 proteins, respectively) due to the high proportion of extracellular matrix (ECM) and correspondingly fewer cells (Fig. 1d). We used the peripheral blood mononuclear cell (PBMC) proteome as the benchmark library for the analysis of skin-associated epidermal and dermal T cells (Fig. 1d).

To visualize proteomic similarities and differences across skin layers and cell types, we employed principal component analysis (PCA) (Fig. 1e). Skin layers and cells segregated by the first and second component with 24.0% and 11.0% of the total data variation, respectively. The separation between skin layers and skin-derived immune cells was partially driven by proteins of the ECM, including collagens and collagen associated proteins of the skin, such as dermatopontin, prolargin, decorin, and lumican. Overall, we found that the majority of proteins (6025; 56.3%) were expressed in all four layers, whereas 3631 (33.9%) were expressed in some, but not all layers, and only few (569; 5.3%) were exclusive for one specific layer (Fig. 2a). These findings are in line with previous work on other tissues showing that the majority of expressed proteins are shared across multiple tissue types while protein abundance varies substantially[21–23].

**The spatial gradient of the skin proteome.** A prominent feature of the skin is its layered composition, with each layer having a distinct functional role. Previous analyses have documented differences that suggested the existence of the molecular spatial gradients of structural and immune proteins across skin layers[1,24–27]. To capture this at the proteomic level, we next analyzed our DDA dataset for the structural and immunological composition of the skin layers as a function of the spatial dimension. Below, we summarize several key observations.

First, our analysis of epidermis identified and quantified 47 keratins (KRTs), which represents all known skin KRTs[24,28]. We found that KRTs 1, 5, 10, 14, and 6A were most abundant in epidermis, comprising 87% of KRTs in the outer epidermis and 93% in the inner epidermis (Fig. 2b, e, g, I, Supplementary Data 2). They make up 15% of the total inner epidermal

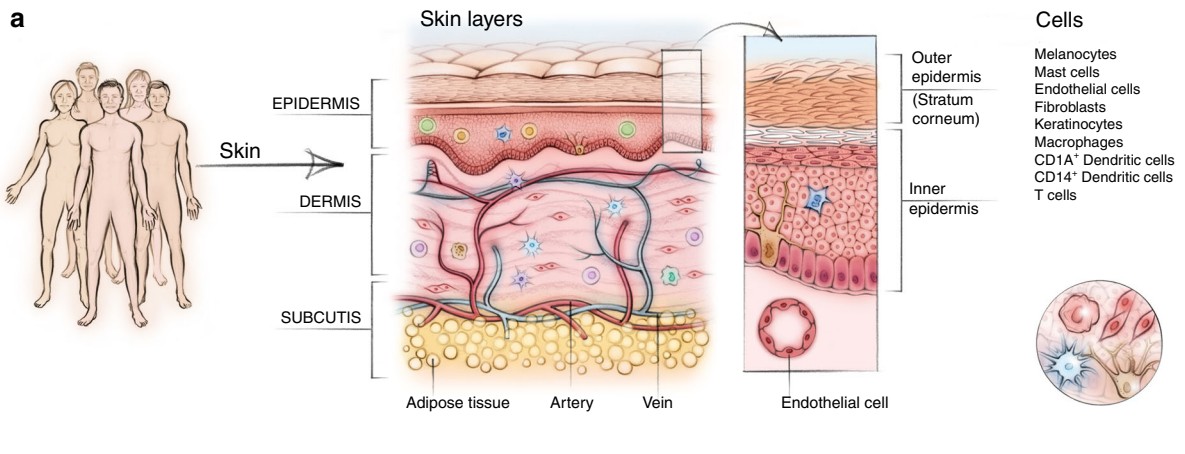

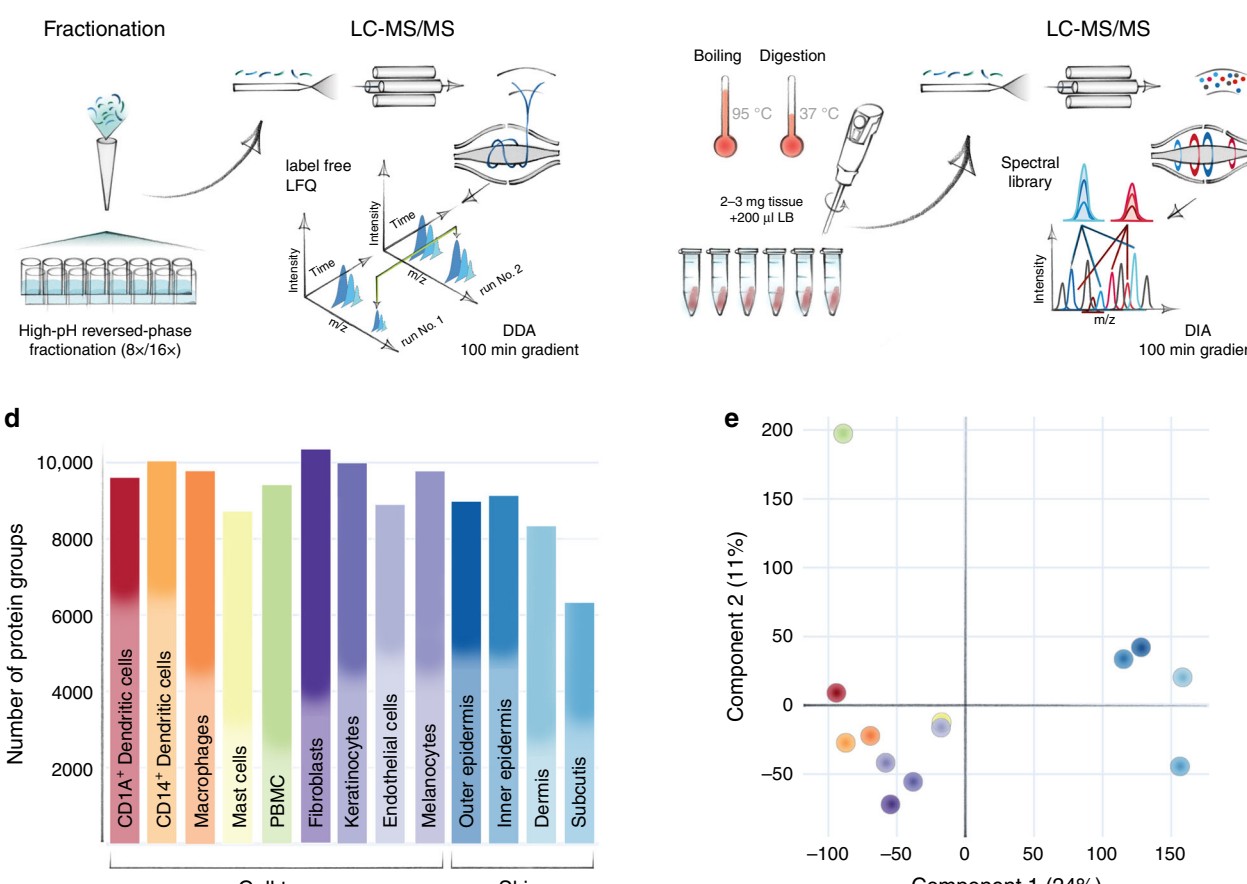

**Fig. 1 In-depth MS-based proteomic analysis of skin layers and its cellular subsets. a** Workflow for measuring the skin proteome including skin layers, cultivated keratinocytes, and fibroblasts and FACS-sorted melanocytes, endothelial cells, and immune cells. **b** High-resolution MS analyses of fractionated, pooled samples in data-dependent acquisition mode (DDA) and subsequent (**c**) single-run analyses of three to six biological replicates in data-independent (DIA) acquisition mode. **d** Number of protein groups for each major cell subset and skin layers based on pooled samples (one independent experiment) for cell subsets and skin layers, respectively examined over one independent experiment (CD1A+dendritic cells ($N = 3$), CD14+ dendritic cells ($N = 3$), macrophages ($N = 4$), mast cells ($N = 6$), fibroblast ($N = 5$), keratinocytes ($N = 5$), endothelial cells ($N = 4$), melanocytes ($N = 5$), skin layers ($N = 5$). A total of 10,701 protein were identified. **e** Principal component analysis (PCA) of all proteomes from skin layers, primary cells and immune cell types. Color code from panel (**d**). The fractionated proteome of PBMC (green dot) was included as library for skin-associated T cells. Component 1 and 2 account for 24 and 11% of the total data variation, respectively.

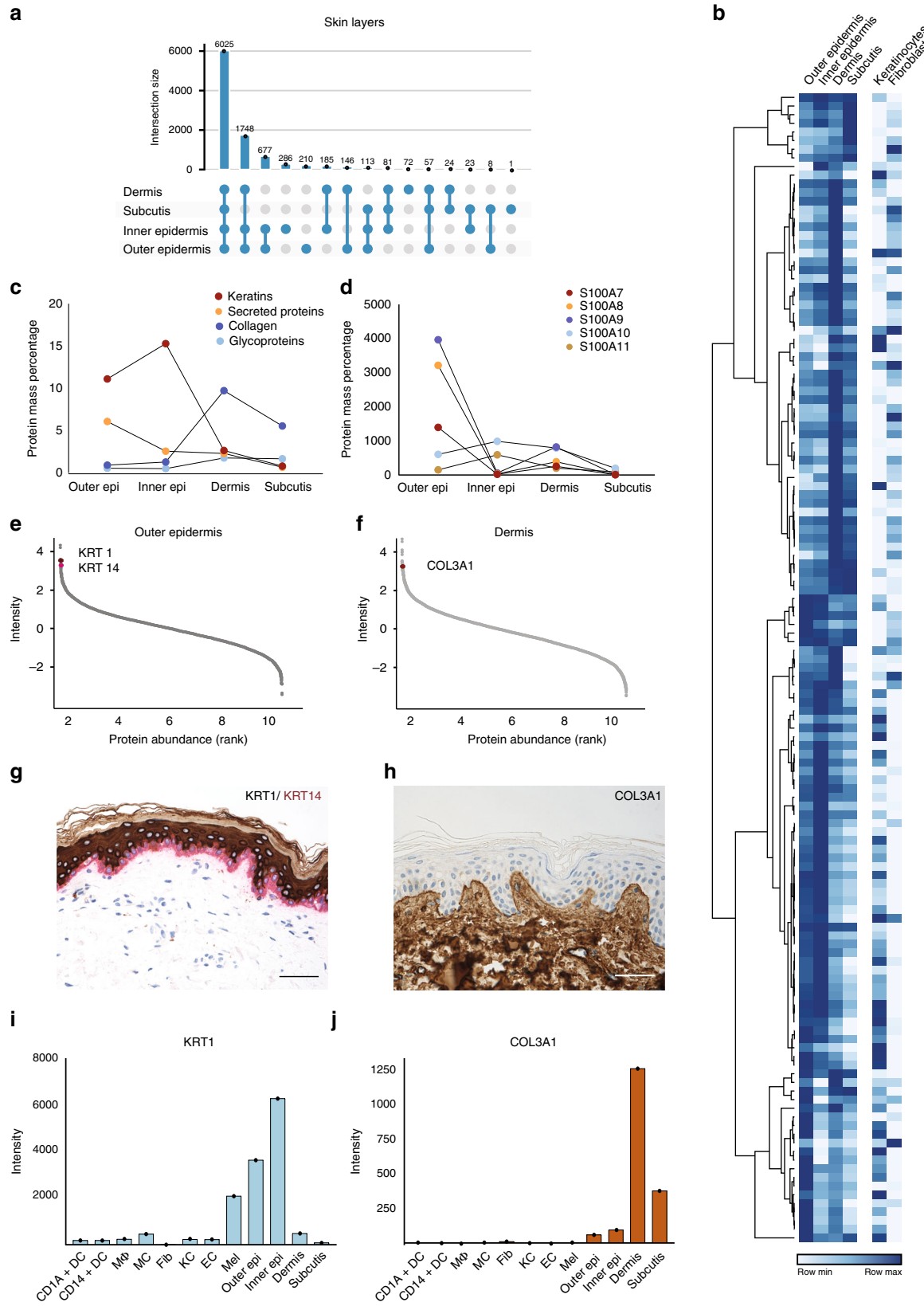

proteome (Fig. 2c). KRT 16 and 17 are ascribed specific roles as "alarmins" of the immune system[29], and supporting this notion, KRT 16 made up 6.7% of total keratins in the outer epidermis but only 0.1% of inner epidermis (2.6% and 0.3% for KRT 17). Moreover, there were clear patterns in abundance of

transglutaminases, enzymes that catalyze crosslinking the proteins of cornified cell envelope to render the outer skin mechanically stable[30]. We observed that transglutaminase 1 and 3 were by far the most abundant in epidermis (0.02%, 0.01% of protein mass, respectively), whereas forms 5 and 2 were

**Fig. 2 Spatial proteomic gradient of human skin. a** Unique and overlapping protein groups in outer and inner epidermis, dermis and subcutis skin layers (UpSet plot) One independent experiment of pooled samples ($N = 1$) from five donors for each layer. **b** Relative expression levels of important structural and immunological proteins across skin layers and in primary keratinocytes and fibroblasts. Clustering is based on the log2 expression level of the proteins in skin layers (DDA). **c** Protein mass percentage of keratins, collagens, glycoproteins and secreted proteins across skin layers. Percentage was calculated by dividing the summed iBAQ intensities of the proteins of interest by all summed iBAQ intensities of all proteins (see Methods). **d** Expression profiles of the S100A family members of proteins across skin layers, illustrating their layer-specific, functional abundance. **e, f** Estimated protein abundance vs protein rank highlighting keratin 1 (KRT1), keratin 14 (KRT14), and collagen III (COL3A1) (upper panels) and (**g, h**) immunohistochemical stainings (KRT1, brown, KRT14, red, COL3A1, brown) (lower panels), scale bar: 50 μm. KRT1 is the protein with the 7th highest estimated abundance in outer epidermis, whereas KRT14 is the 6th most abundant. COL3A1 is the 19th most abundant protein in dermis, which is also reflected in the staining. **i** Estimated abundance of KRT1 and (**j**) COL3A1 across libraries based on pooled samples of skin layers and skin-derived cells examined over one independent experiment from pooled samples ($N$ = number of pooled donor samples). (CD1A[+]dendritic cells ($N = 3$), CD14+ dendritic cells ($N = 3$), MΦ; macrophages ($N = 4$), MC; mast cells ($N = 6$), Fib; fibroblast ($N = 5$), KC; keratinocytes ($N = 5$), EC; endothelial cells ($N = 4$), Mel; melanocytes ($N = 5$), skin layers (Outer Epi, Inner Epi, Dermis, Subcutis) ($N = 5$). Source data are provided as a Source Data file.

comparatively ten times less abundant (Fig. 2b). We found distinct gradients in their expression suggesting that this crucial function happens mainly in the epidermis (Fig. 2b, Supplementary Data 2). In another example, our data show that the desmosomal proteins desmoglein, desmocollins, plakoglobin, desmoplakin, envoplakin, periplakin and plakophilin alone account for approximately 1.8% of protein mass of the inner epidermis (Fig. 2b). This is in agreement with the critical role of desmosomes in forming intercellular junctions that link the keratinocytes to each other and to the intracellular filament (IF) network, thus resulting in adhesive protein network that functions to withstand mechanical stress in the basal layers of epidermis[31].

Moving towards deeper skin layers, we found that glycoproteins such as elastin and laminins were the most abundant in the dermis (Fig. 2b, c), with elastin accounting for 20% of the annotated glycoproteins in the dermis. Of the collagens, we found all 16 that have been described in interfollicular skin before, with collagen I, III, IV, and VI alone comprising 96% of their total mass in the dermis (Fig. 2b, f, h, j, Supplementary Data 2). Interestingly, we quantified low amounts of collagen XXI (COLXXI), a family member that has so far only been described in extracellular matrix in the heart, stomach, kidney, placenta and smooth muscle[32]. Together collagens make up 10% of total protein mass in the dermis (Fig. 2c). Elastins, laminins and collagens are thought to provide structural scaffolding and elasticity to the dermis; our quantitative proteomic data confirms that these proteins are well placed to carry out these functions. In addition to its structural role, the skin acts as both an active and passive immunologic barrier. Therefore, we next analyzed the spatial gradient of proteins with immunological roles. The members of the S100 family of $Ca^{2+}$- and $Zn^{2+}$ binding proteins were by far the most abundant, and their members had very distinct spatial profiles. Whereas S100A7, S100A8, and S100A9 were more abundant in outer epidermis, S100A14/16 were more abundant in epidermis and keratinocytes, and S100A6 in dermis (Fig. 2b, d, Supplementary Data 2). These observations agree with the proposed roles of S100 proteins in pro-inflammatory and antimicrobial activities and keratinocyte differentiation, and the previously reported differences in their expression profiles[26].

Our proteomic data also demonstrated the presence of a wide arsenal of immunologically active proteins, including chemokines (CXCL12/14), antimicrobial peptides (defensin β1/β2), matrix metalloproteinases (MMP2/3/9) and cytokines. We next analyzed expression of the members of the interleukin (IL) 1 cytokine family; interleukins play critical roles in defending against infection and contribute to inflammatory skin diseases[33,34]. Interleukins are extremely difficult to identify with proteomic methods because of their low abundance, especially in the absence of inflammation. Remarkably, we identified all members of the

IL-1 family and quantified their spatial and cellular expression patterns, including IL-18, IL-37, IL-38 and IL-36γ, which were mainly detected in the epidermal layers (Fig. 3). Unexpectedly, IL-36α, a poorly described family member, was restricted to the subcutis, and at least nine fold more abundant than other IL-1 family members in any skin compartment (Fig. 3f). Low levels of the recently described cytokine IL-33 were localized to the dermis, and further localized to endothelial cells, as reported in the literature (Fig. 3g, h)[35]. IL-1β was readily quantified in isolated dendritic cells and macrophages. We did not detect IL-1β protein in healthy skin by immunohistochemistry, but IL-1β was produced by macrophages and dendritic cells from inflamed skin (Fig. 3i, Supplementary Fig. 3). Lastly, medically targeted cytokines known to drive inflammatory skin diseases such as psoriasis (IL-17, IL-22, and IL-23) and atopic dermatitis (IL-4, IL-5, IL-13) were not identified in healthy skin. Taken together, these studies represent a detailed quantitative proteomic analyses of immunologically active proteins within the distinct structural compartments and cell types in human skin.

**Proteomic atlas of cultivated fibroblasts and keratinocytes.** Following our proteomic characterization of skin layers by data-dependent acquisition, we next analyzed the constituent cell types in skin. Keratinocytes and fibroblasts are by far the most frequent component cell types, followed by the endothelial cells that line blood and lymphatic vessels, pigment-producing melanocytes and then the cellular subsets of the resident innate and adaptive immune systems. We chose DIA as the best method to compare proteomes between these diverse cell types. To minimally disturb the proteomes, we isolated and FACS-sorted cells directly from single-cell suspensions of digested skin. However, this was not appropriate for fibroblasts and keratinocytes because both lack distinct and stable surface markers for sorting. We therefore used cultured fibroblasts and keratinocytes to characterize the proteome and to identify novel and specific cell type markers.

Endosialin (CD248/TEM1) has previously been reported to be a intracellularly expressed protein in skin fibroblasts[36]. We therefore investigated its MS-based abundance across cell types. Endosialin was within the top third of the proteome in fibroblasts, whereas it was hardly detected in any other cell type. Thus, we used the endosialin abundance profile across cell types as a reference to identify the top 100 proteins with similar abundance pattern (Fig. 4a, b, Supplementary Data 3). Some of the proteins identified using this strategy include SLIT2, FGF2, PRRX1, which have been mentioned in the context of fibroblast biology[37,38], LOXL4, with a role in collagen crosslinking[39], and GREM1, COL11A1, LRRC15, all differentially associated with cancer-associated fibroblasts but not quiescent skin fibroblasts[40,41] (Fig. 4c, d). The analysis also highlighted the protein C12orf75 that, to the best of our knowledge, has not been characterized in

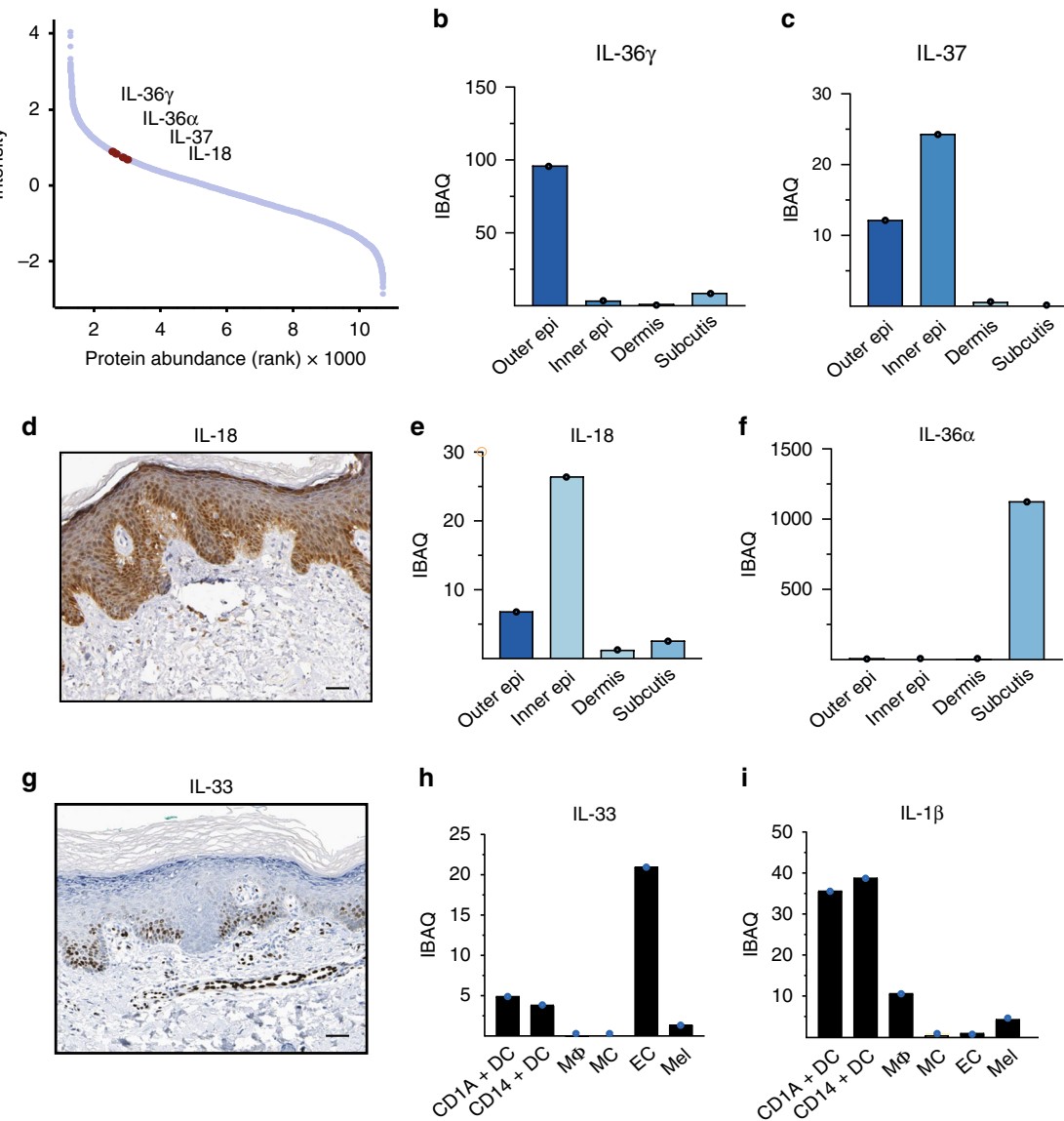

**Fig. 3 Expression levels of interleukin 1 family members in the skin. a** Dynamic range plot illustrating the abundance of interleukin (IL) 1 family members in the skin. **b**, **c** Estimated intensity of the IL-1 family members IL-36γ and IL-37 throughout the skin. IL-36γ is primarily found to be expressed in the outer epidermal layer whereas IL-37 is both found to be expressed in the outer- and inner epidermal layer. **d** Immunohistochemical staining of healthy skin obtained from the Human Protein Atlas (Methods, Supplementary Data 17) showing IL-18 protein expression primarily in the inner epidermal layer (dark brown), scale bar: 50 μm. **e**, **f** Estimated intensity of the IL-1 family members IL-18 and IL-36α throughout the skin. IL-18 is mainly found in the inner- and outer epidermis whereas IL-36α is found in subcutis. **g** Immunohistochemical staining of IL-33 expression on healthy skin obtained from the Human Protein Atlas (Methods, Supplementary Data 17), scale bar: 50 μm. **h** Estimated intensity of recently discovered IL-33, here depicted in FACS-sorted skin-derived endothelial cells (EC), and to lesser extent in macrophages (MΦ) and dendritic cells (DC). **i** Estimated intensity of IL-1β expression in FACS-sorted cellular subsets from healthy skin. All barplots show estimated intensity (IBAQ, log2) based on one library of pooled samples (N = number of pooled samples) of skin layers (N = 5; **b**, **c**, **e**, **f**), CD1A+dendritic cells (N = 3; h, i), CD14+ dendritic cells (N = 3; h, i), MΦ; macrophages (N = 4; **h**, **i**), mast cells (N = 6; **h**, **i**), EC; endothelial cells (N = 4; **h**, **i**), Mel; melanocytes (N = 5; **h**, **i**). Source data are provided as a Source Data file.

fibroblasts, but has been found overexpressed in colon cancer (Fig. 4e)[42].

A similar analysis on the primary keratinocytes revealed that the known marker kallikrein 10 was in the top half of the quantified keratinocytes proteome, with negligible expression elsewhere (Fig. 4f, Supplementary Data 4).

The top 100 proteins that best correlated with keratinocytes included cell-to-cell adhesion proteins (cornifelin, plakophilin 3, TNS4, DSG3, LAD1), proteins associated with epithelial cell transformation and differentiation (FAM83B, ZNF185) and KLK5, an additional keratinocyte-specific kallikrein[6,43–48]. Our analysis also uncovered a number of proteins that have not

been functionally characterized in the context of keratinocyte biology, including TMEM40, FGFBP, CRCT1, EGFLAM, Anxa8AL1, and NT5DC3 (Fig. 4g–j). Conversely, a number of functionally important proteins that have been documented as expressed by keratinocytes during the differentiation process in vivo were not identified in monolayer keratinocyte cultures. Among these are a number of keratins, late cornified envelop proteins such as LCE1E, LCE1F, and LCE6A and loricrin. As we had readily identified almost all of these proteins in the relevant skin layers, these differences likely exemplify the biological differences between cultured and in vivo keratinocytes.

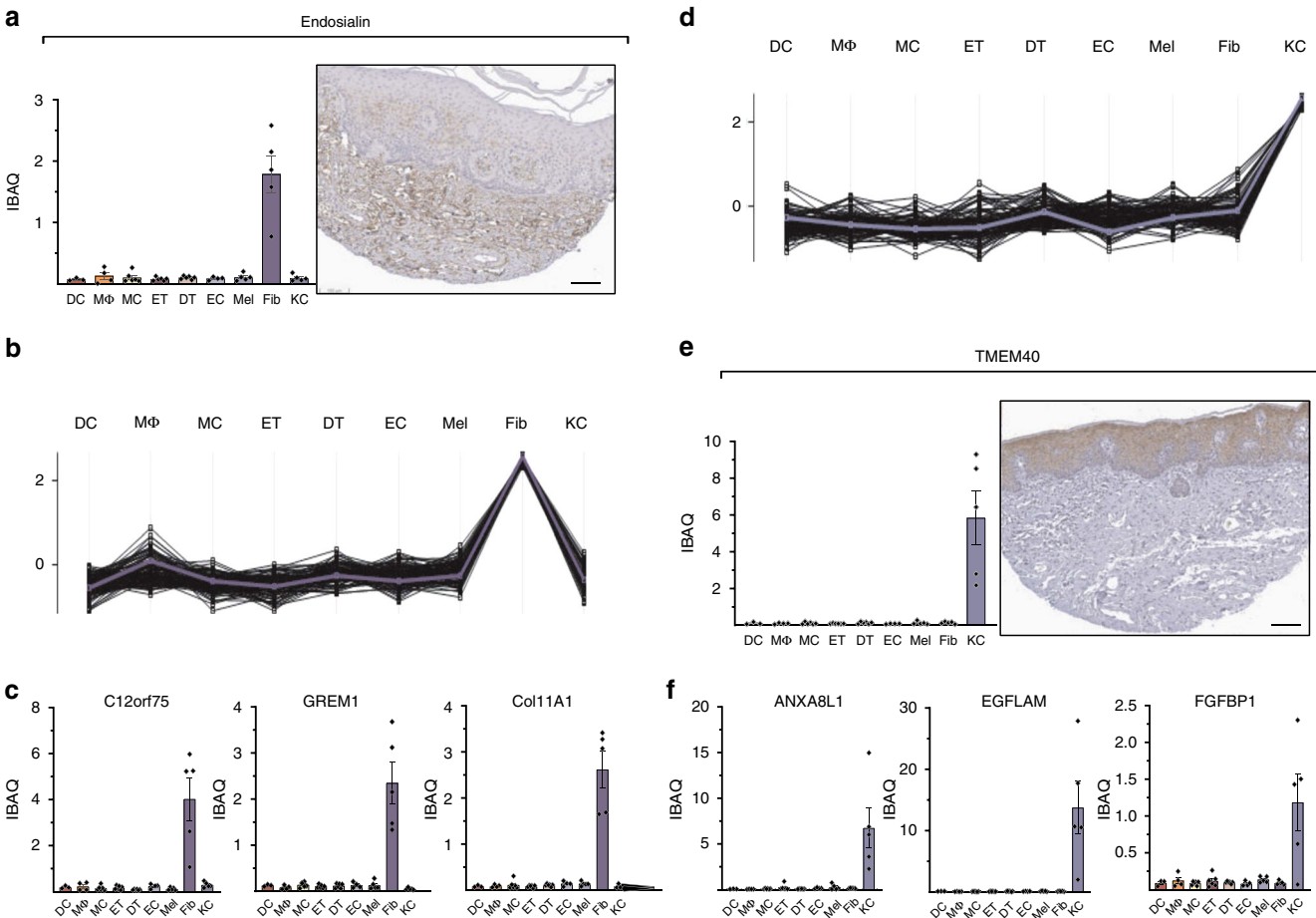

**Fig. 4 The proteomes of cultivated fibroblasts and keratinocytes. a** MS-intensity (iBAQ) of endosialin across cell types (left upper panel) compared to its immunohistochemical staining obtained from the Human Protein Atlas (Scale bar: 100 μm, see Methods and Supplementary Data 17). **b** Protein abundance profiles of the top 100 proteins across cell types with the most similar fibroblast-enriched profile as endosialin (reference profile, blue line). Each line represents one protein. **c** MS-intensity of selected fibroblast-enriched proteins (GREM1, Col11A1, c12orf75,) across cell types. **d** Protein expression profiles of the top 100 proteins across cell types with the most similar keratinocyte-enriched profile as KLK10 (blue line). **e** iBAQ values of the keratinocyte-enriched protein TMEM40 across cell types and its immunohistochemical staining obtained from the Human Protein Atlas (Scale bar: 100 μm, see Methods and Supplementary Table 17). **f** MS-intensity of selected keratinocyte-enriched proteins (ANXA8L1, EGFLAM, FGFBP1) across cell types. All barplots show estimated intensity (data are presented as mean±SEM (IBAQ, log2)). DC; CD1A$^+$dendritic cells ($N = 3$), MΦ; macrophages ($N = 4$), MC; mast cells ($N = 6$), DT; dermal T cells ($N = 6$), ET; epidermal T cells ($N = 6$), EC; endothelial cells ($N = 4$), Mel; melanocytes ($N = 5$), Fib; fibroblasts ($N = 5$), KC; keratinocytes ($N = 5$). Source data are provided as a Source Data file.

**The proteomes of cellular subsets reveal unexplored proteins.** Comprehensive proteomic analyses of the cellular subsets of human skin has been hampered by the technical challenges of obtaining the number of cells needed for these analyses. Consequently, the majority of studies published to date relied on either immunohistochemical analyses or on cultured cells. We prepared freshly isolated single-cell suspensions from the skin of three to six healthy donors and used multi-parameter flow-cytometry based sorting to isolate purified cell types. We prepared 34 samples that were analyzed by single-shot DIA MS-based proteomics (Fig. 5) (see "Methods"). In addition to the endothelial cells and melanocytes mentioned above, we analyzed mast cells, macrophages and dendritic cells of the innate immune system and epidermal and dermal T cells of the adaptive arm of the immune system. We used the deep proteomes of cellular subsets including endothelial cells, melanocytes, dendritic cells, mast cells, macrophages and PBMCs as matching libraries for DIA (constituting between 8726 and 10,337 proteins, Fig. 1d). To verify the high purity of the FACS-sorted cell types, we employed the MS-signals of the characteristic markers of all cellular subsets (Supplementary Fig. 1d–m). For each of the FACS-sorted cell types, we

covered a quantitative proteome of between 5100 and 6700 proteins (Fig. 5a). We visualized commonalities and differences between cellular subsets with a PCA plot, in which we also included the cultivated subsets (Fig. 5b).

As expected, cultured subsets clustered away from freshly isolated subsets; proteins associated with cornification and keratinization driving the separation of keratinocytes and functionally important proteins such as CPA3, CMA1, and CDH5 contributing to the separation of immune cells. Within the FACS-sorted cell types, melanocytes and endothelial cells clustered closely together and apart from immune cells, despite having very distinct roles in the skin. The two different T-cell populations were also substantially separated (Fig. 5d). Standard ANOVA analysis between cellular proteomes of the FACS-sorted cells revealed large differences in the expression profiles of these populations (6713 of a total of 8212 proteins, FDR < 0.05, Supplementary Data 5). Gene ontology (GO) enrichment analysis in the first instance highlighted differences between the melanocytes and endothelial cells vs. immunological cell types. For the melanocytes, we observed enrichment of proteins involved in melanin biosynthetic process (KIT, MLANA,

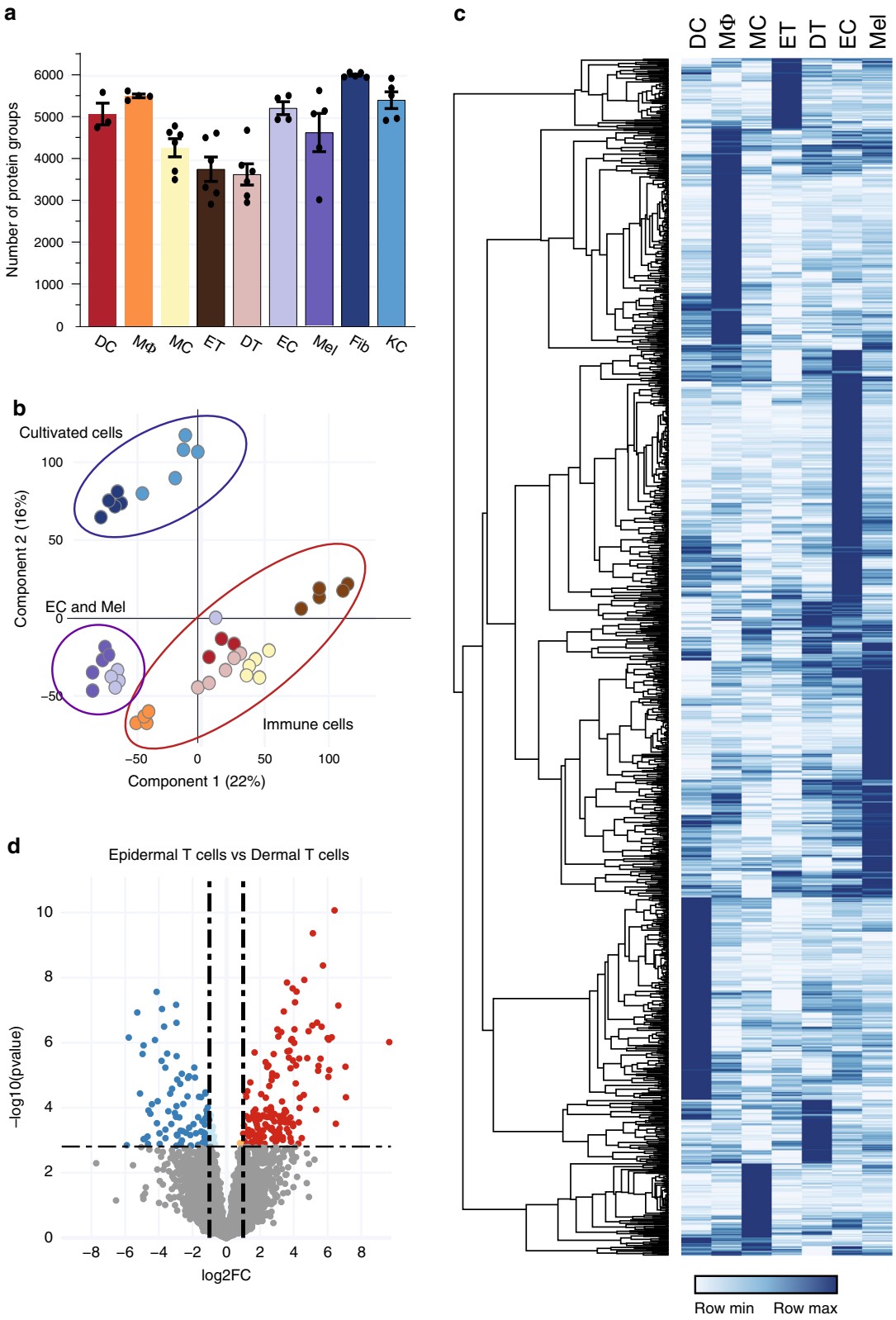

TYRP1, DCT, DDT, TYR, and MYO5A) as well as phagosome acidification-associated proteins responsible for dispersing melanin to neighboring keratinocytes[49]. We next ran a posthoc pairwise t-test analysis across all cell types to reveal proteins that were significantly different in at least two cell types (FDR < 0.01; fold-change>2). This stringent approach revealed a set of 1272 such proteins, including proteins involved in Toll-like

receptor (TLR) signaling pathway in macrophages and proteins involved in antigen processing in dendritic cells. Hierarchical clustering of these proteins based on abundance levels across cell types yielded a heatmap with clearly distinct protein clusters (Fig. 5c; Supplementary Data 6). Within each of these clusters, in addition to proteins with the established functions in the respective cell types, we observed proteins without a

**Fig. 5 In-depth MS-based proteomic analysis of skin-associated cellular subsets. a** Single-run analyses of DC; CD1A$^+$dendritic cells ($N = 3$), MΦ; macrophages ($N = 4$), MC; mast cells ($N = 6$), ET; epidermal T cells ($N = 6$), DT; dermal T cells ($N = 6$), EC; endothelial cells ($N = 4$), Mel; melanocytes ($N = 5$), Fib; fibroblasts ($N = 5$) and KC; keratinocytes ($N = 5$) using data-independent (DIA) acquisition. Data are presented as mean±SEM. $N$ represents number of biologically independent samples. The number of quantified protein groups for each major cell lineage is roughly similar. Source data are provided as a Source Data file. **b** Principal component analysis (PCA) of all proteomes from cellular subsets. Color code from panel (**a**). The PCA separates cultivated fibroblast and keratinocytes from FACS-sorted endothelial cells (EC) and melanocytes (Mel) as well as from the immune cells, as indicated by enclosing ovals. **c** Heatmap of protein abundances of 1272 differentially expressed proteins (ANOVA, FDR < 0.01, FCH > 2) after unsupervised hierarchical clustering. **d** Differentially expressed proteins in epidermal T cells vs. dermal T cells (volcano plot, FDR < 0.05, FCH > 2).

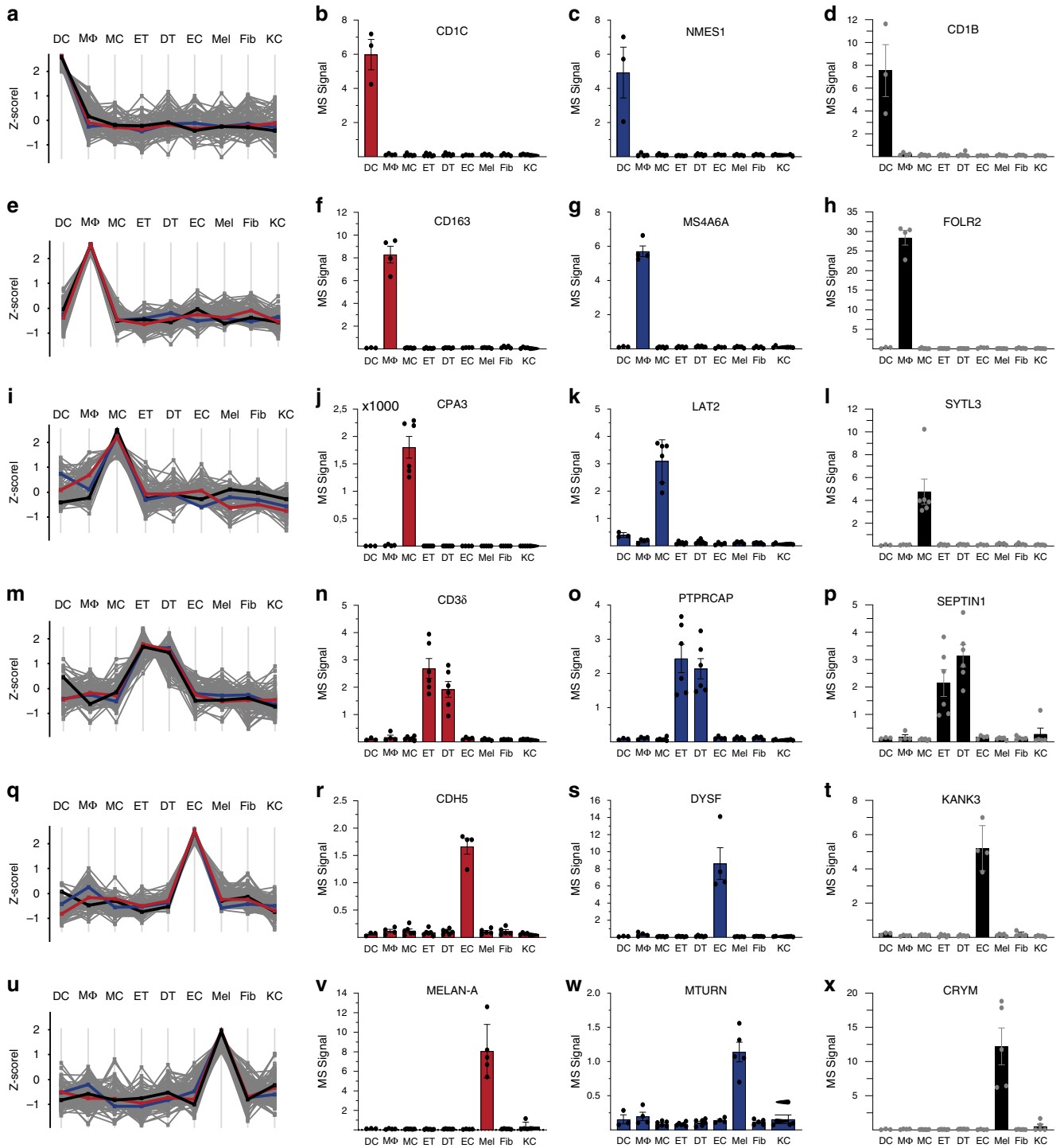

**Fig. 6 The proteomes of cellular subsets from the skin reveal subset enriched proteins.** Protein expression profiles for the top 100 proteins across cell types using specific, well-characterized proteins as reference for each subset. **a** Protein expression profile of DCs using (**b**) CD1c as a reference (red line in (**a**)). Protein expression of selected proteins enriched in DCs (**c**) NMES1 and (**d**) CD1b across cell types. **e** Protein expression profile of MΦ using (**f**) CD163 as a reference profile (red line in (**e**)). Protein expression of selected proteins enriched in MΦ (**g**) MS4A6A and (**h**) FOLR2 across cell types. **i** Protein expression profile of MCs using (**j**) carboxypeptidase A (CPA3) as a reference profile (red line in (**i**)) and expression of selected proteins enriched in MCs (**k**) LAT2 and (**l**) SYTL3) across cell types. **m** Protein expression profile of ETs and DTs using (**n**) T-cell receptor chain CD3δ as a reference profile (red line in (**m**)). Protein expression of selected proteins enriched in ETs and DTs (**o**) PTPRCAP and (**p**) SEPTIN1 across cell types. **q** Protein expression profile of ECs using (**r**) adhesion molecule cadherin 5 (CDH5) as a reference profile (red line in (**q**)) and bar plots illustrating the expression of selected proteins enriched in ECs (**s**) DYSF and (**t**) KANK3 across cell types. **u** Protein expression profile of melanocytes (Mel) using (**v**) adhesion molecule melanogenesis pathway member Melan-A (MELAN-A) as a reference profile (red line in (**u**)) and bar plots illustrating the expression of selected proteins enriched in melanocytes (**w**) MTURN and (**x**) CRYM across cell types. All barplots are presented as mean±SEM, (log2). DC; CD1A[+]dendritic cells (N = 3), MΦ; macrophages (N = 4), MC; mast cells (N = 6), DT; dermal T cells (N = 6), ET; epidermal T cells (N = 6), EC; endothelial cells (N = 4), Mel; melanocytes (N = 5), Fib; fibroblasts (N = 5), KC; keratinocytes (N = 5). N represents number of biologically independent samples. Source data are provided as a Source Data file.

well-established role in the given cell type. This substantial group contained 39 kinases and 16 ubiquitin protein ligases whose function in skin biology has not been established (Supplementary Data 7 and 8).

Next, we performed a profile analysis on the FACS-sorted cells. For each cell type, we chose previously characterized proteins with cell type restricted expression profiles as references and determined the top 100 proteins with the most similar profiles (Fig. 6). We observed that although expression of reference proteins was very restricted to the respective cell type, profiles of the top 100 proteins displayed more variability, indicating that there is a limited number of proteins with unique expression. This was particularly the case for mast cells and melanocytes. The distinct T-cell profile resulted from combining dermal and epidermal T cell populations (for distinguishing proteins see below). Together, these characteristic sets that emerged from the hierarchical clustering and the profile analysis may represent proteins that are functionally relevant for these skin cell types, and therefore their future study could reveal novel aspects of their biology. Below, we highlight some of the interesting observations that emerged from our initial analysis of characteristic sets for dendritic cells, macrophages, mast cells, skin T cells, endothelial cells, and melanocytes.

Dendritic cells act as sentinels by detecting pathogens in the skin, and serve as a link between the innate and the adaptive arms of the immune system. These cells have characteristic expression of C-type lectin receptors that recognize glycan structures on pathogens and commensals, and CD1a, CD1b, and CD1c molecules, that present lipid and glycolipid antigens to other immune cells[50]. Using CD1c as the reference, we selected 99 proteins with the most similar expression pattern (Fig. 6a, b, Supplementary Data 9). This characteristic set included, as expected, CD1b (Fig. 6d), as well as several C-type lectin receptors (CLEC4A, CLEC5A, CLEC10A), activation marker (TRAF1), chemokine (CCL22) and chemokine receptor (CCR2), and immune function (PLD4, SIGLEC9, ALOX15. We also detected NMES1 (Fig. 6c), a protein recently described to be specific for dendritic cells and involved in effector functions[51,52]. Interestingly, many of the proteins that are specifically enriched in dendritic cells, for example enzymes such as PDCL3 and FCHO1 have no known function in the skin and represent strong candidates for follow up studies.

To define the macrophage "characteristic set" we used the innate immune sensor CD163 as a reference protein (Fig. 6e, f, Supplementary Fig. 1, Supplementary Data 10). It included proteins with known functions in innate immunity (LILRB4, C1QA-C, C3AR1, CD14, C2, C8A), chemotaxis (CCL18, CCL3), immune regulatory function (ABC1, VSIG4) and stress response in the context of tumorigenesis (RNASE2, GPX3), as well as

FOLR2 (Fig. 6h). This set of proteins also included MS4A6A and TMEM176B, members of the MS4A family, whose founding member is CD20 (MS4A), a well-characterized B-cell marker. Both MS4A6A and TMEM176B have not been functionally explored but are highly enriched in macrophages (Fig. 6g)[53,54].

For analysis of mast cells, we used carboxypeptidase A (CPA3) as reference (Fig. 6i, j, Supplementary Data 11). The resulting characteristic set of proteins selectively enriched in mast cells includes regulators of mast cell activation (RGS13, PIK3R6, PIK3C2B, RHOG), characteristic proteases (tryptase, chymase and carboxypeptidase M), proteins involved in innate immunity (TRIM38, TRIM23) and proteins with unknown function in the context of mast cell biology (LAT2, SLP3) (Fig. 6k, l).

Skin T cells are dispersed in the skin and mostly prevalent in the dermis in proximity to blood vessels, hair follicles and eccrine sweat ducts[4]. There are different populations of skin T cells, most prominently dermal and epidermal T cells, which are all defined by the expression of a T-cell receptor (TCR) that is assembled from multiple chains. We used the CD3δ chain of TCR to define characteristic set of both dermal and epidermal T cells (Fig. 6m, n, Supplementary Data 12). Apart from the other constituents of the TCR, including CD3γ and CD3ε chain, the known surface receptors CD5 and CD6 were also exclusively expressed by T cells. Other proteins involved in regulation of TCR stimulation (SKAP1, SH2D1A, SIT1, FYB1, CARMIL2, LIME1, PTPRCAP), cytolysis (GZMA, GZMK, CTSW), antiviral activity (APO-BEC3G), complement activation (C5) and cytokine production (KAMK4, IKZF3, GBP5) were found to be highly enriched in T cells. Remarkably, the T-cell profile set also contained many proteins that are completely uncharacterized or remain to be functionally characterized with respect to their function in T cells (PYHIN1, ACAP1, PNPLA7, SEPTIN1) (Fig. 6o, p). We also observed that epidermal and dermal T cells exhibited clear differences (Figs. 5b and 6d, Supplementary Data 13), further supporting the proteomic diversity we observed in distinct skin layers.

Endothelial cells line the blood and lymphatic vessels of the skin. These cells act as a semi-selective barrier between the lumen and the surrounding tissue and regulate trans-endothelial migration of immune cells. We used cadherin 5 (CDH5), an adhesion molecule between endothelial cells, as a reference for defining the characteristic set of endothelial cells (Fig. 6q, r, Supplementary Data 14). The resulting set included typical endothelial cell proteins such as leukocyte adhesion molecule selectin E and P (SELE and SELP, respectively), tight junction protein claudin 5 (CLDN5), and endomucin (EMCN), a regulator of VEGF mediated cell migration and growth. The set also confirmed endothelial cell enrichment of proteins such as RAPGEF3 and ABCB1, regulators of vascular permeability[55]

and the C-type lectin CLEC14A[56]. Furthermore, our analysis yielded proteins such as KANK3, DYSF, HSPA12B, MYCT1 that have not been functionally unexplored in the context of human endothelial biology (Fig. 6s, t).

We used Melan-A (MLANA) as a characteristic marker for melanocytes (Fig. 6u, v, Supplementary Data 15), the major cellular source of the UV protective pigment melanin[49]. The resulting set grouped together members of the melanogenesis pathway such as tyrosinase (TYR), dopachrome tautomerase (DCT) and neural crest-derived cell markers S100 and Cellular retinoic acid binding protein 1 (CRABP1) (Supplementary Fig 1). However, the analysis also identified a SLC6 family transporter (SLC6A17), proteins that are enriched in brain and neural crest-derived cells but lack biological characterization in the skin. We also identified several proteins with no known role in melanocyte biology, including SYNGR1, CRYM, MTURN, MAPT, MAP2, FBLL1 (Fig. 6w, x).

Taken together, the integration of FACS cell sorting, MS-based quantitative proteomics, and protein marker based analyses of cell type-specific characteristic sets revealed a large number of proteins with unexplored roles in skin biology.

## Discussion

The skin defines the interface of the human body with its external surroundings. It has a variety of functions that are reflected in its intricate cellular and mechanical structure. These functions are carried out by proteins and yet, to date there has been no in-depth, cell-type- and location-specific proteomic studies of this organ. The high dynamic range (difference between most abundant and least abundant proteins) of the skin proteome have long presented an unsurmountable barrier to comprehensive quantitative proteomic analyses. The presence of highly abundant proteins, such as collagens in the dermis, make it difficult to detect low abundance proteins. Additionally, many cell types are present in low numbers and cannot be effectively characterized with current levels of sensitivity.

In this study, we overcame these challenges and quantitatively characterized the proteomic composition of healthy human skin. We used a combination of curettage, skin dissection, FACS sorting and primary cell culture to separate the skin into four layers and nine cell types. We successfully generated a proteomic atlas of 10,701 proteins, by far the largest collection of proteomic data obtained to date from human skin. We expect that many novel and unexpected insights will emerge from future analysis of this data, given its comprehensive and multidimensionally resolved nature.

We provide some initial illustrations of the types of insights that our dataset can yield. For example, our structural protein quantification revealed that just five of all the described keratins in epidermis—KRT 1, 5, 10, 14, and 6A—accounted for 87% of total keratin mass and 10% of total protein mass in epidermal compartments, suggesting that low abundance keratins may have non-structural roles (such as serving as alarmins)[29]. Likewise we quantified 16 collagens in dermis and subcutis, including collagens I, III, IV, and VI, which accounted for 96% of the total collagen mass in the dermis. We also made the observation that collagen XXI is expressed in human skin, which has not previously been described. Knowledge of the identity, quantity, and spatial distribution of these key skin proteins will contribute to a more holistic understanding of skin structure and function.

To characterize the proteins underlying the immune-protective role of the skin, we quantified a large number of proteins with pro-inflammatory- and antimicrobial activities, including chemokines, matrix metalloproteinases, S100 family members, interleukins and other cytokines. These studies revealed that there were surprisingly large quantitative and spatial differences in protein expression in different layers of the skin. We also comprehensively cataloged the protein composition of cultured first passage keratinocytes and fibroblasts to complement our skin layer-based proteomic analyses. Lastly, we further captured the complexity of the human skin proteome by separately analyzing major constituent cell types, including melanocytes, endothelial cells, macrophages, dendritic cells, mast cells, and dermal- and epidermal T cells. Our analyses identified distinct proteomic profiles in each cell type, indicating that unique group of protein defines their differential biology. Strikingly, we identified proteins in each cell type that have not been previously associated with skin biology. We predict that further mining of this proteome atlas will expand our fundamental insight into the biology of skin.

In summary, we describe a high-resolution comprehensive proteomic analyses of healthy human skin, including the detailed study of different skin layers and distinct cell types. It is our hope that this atlas will serve as a rich resource for future analyses of skin and immune cell function and serve as a foundation for the study of proteomic alterations in skin disease. Future proteomic analyses of skin disease such as psoriasis could open a door for personalized medicine, identify previously unsuspected pathologic mechanisms, and identify possible new targets for treatment. This atlas could also serve as a rich resource for understanding how single gene-based disorders, such as filaggrin deficiency, affect the surrounding skin and contribute to clinically evident disease. The high sensitivity of our approaches and the and minimal required starting materials should enable in-depth proteomics analyses of even small skin biopsies, perhaps enabling the use of MS-based studies as diagnostic tools.

## Methods

**Ethics oversight**. The study was carried out in agreement with the Danish and Dutch law (Medical Research Involving Human Subjects Act), in accordance with the guidelines of the Medical Ethics Review Committees of the involved institutes and following the Declaration of Helsinki principles.

Discarded skin tissues from corrective surgery of the breast or abdomen were anonymized prior to providing them to the authors of the paper, who were not involved in the surgeries during which the tissues were collected. According to the Danish and Dutch law, researchers are allowed to use discarded anonymous tissue without patient consent.

Human peripheral blood was obtained from healthy adult volunteers (independent from the skin donors) via a vene puncture after written informed consent using an approved protocol by the Ethics Committee of the Capital Region of Denmark (H-3-2014-123 version 1 from 30 07 2018) and complying with the ethical regulations for work with human participants.

**Tissue preparation**. Discarded abdominal skin was obtained from cutaneous surgeries and processed within an hour after surgery. Stratum corneum was curetted carefully with an 8 mm curette and snap frozen in liquid nitrogen. Next, stratum spinosum, stratum granulosum and stratum basale (together with remaining traces of stratum corneum) were curetted and collected as one sample. A 4 mm punch biopsy was collected and divided in upper dermis and subcutis. Visible fat was separated from the rest and snap frozen individually. After each step, a punch biopsy was collected to ensure the histological accuracy (Supplementary Fig. 1a–c). Two additional 4 mm biopsies were collected, one biopsy was covered in Tissue-Tek O.C.T. Compound (Sakura, Cat. no. 4583) and snap frozen and the second biopsy was stored in formaldehyde for 24 h and subsequently paraffin embedded.

Skin biopsies from discarded tissue were incubated in 50 U/ml dispase (Sigma, cat. no. 4942078001) at 4 °C overnight. Epidermis and dermis were separated with forceps. Primary cultures of dermal fibroblasts were initiated by placing 3 mm pieces of the dermal portions in 6-well plates in 3 ml DMEM/F12 fibroblast medium (Corning/Mediatech, cat. no. 10-090-CV) containing 15% FCS (Sigma, cat. no. F7524), 1% penicillin-streptomycin (Sigma, cat. no. 0781), 1% L-Glutamine (Sigma, cat. no. G03202) and EGF (10 ng/mL) (Peprotech, cat. no. AF-100-15). The skin pieces were removed when adherent fibroblasts were visible in the wells. Epidermal strips were pelleted by spinning 250g, 5 min and resuspended in prewarmed trypsin/EDTA (Sigma, cat. no. T3924) and incubated 5 min at 37 °C. Trypsin was neutralized by FCS and cells were rinsed 3 times with RPMI (Corning/Mediatech, Cat. no. 10-040-CV) containing 1% BSA (Sigma, cat. no. 05470) and 1% penicillin-streptomycin. Larger pieces were removed with forceps and remaining cells were counted with trypan blue viability dye (Bio-Rad, cat. no.

1450021) and plated in keratinocyte growth medium (Gibco/Life Technologies, cat. no. 17005-042) including 1% penicillin–streptomycin in 8 cm plates. Cultures were fed three times per week by careful aspiration of approximately half of the culture medium and replacement with fresh medium.

**Processing of skin and purification of skin resident immune cell subsets.** Residual human skin tissue was obtained from healthy donors (aged between 18 and 65 years) undergoing corrective surgery of the breast or abdomen. Within 2 h after resection, sheets of 0.3–0.4 mm thickness were prepared with an electro-dermatome and were treated overnight at 4 °C with 0.2% (wt/vol) dispase II (Boehringer Mannheim, Mannheim, Germany) in PBS with 50 μg/ml gentamycin (Sigma, St. Louis, MO). Epidermis and dermis were separated by forceps, and in order to get fresh single-cell suspensions, the epidermis was fragmented by scissors and incubated in 0.25% (wt/vol) trypsin in PBS for 30 min at 37 °C, while the fragmented dermis was incubated in IMDM (Corning, cat. no. MT10016CV) with 0.4%(wt/vol) collagenase D (Sigma, cat. no. C0130-1G), 50 U/mL DNAse I (Sigma, cat. no. D4263-1VL) and 0.5% (vol/vol) FCS for 2–3 h at 37 °C. After enzymatic digestion, the cell suspensions were sieved through 70-μm cell strainers (Falcon) to remove tissue debris, yielding single cells with a viability exceeding 97%. In order to sort epidermal T cells and melanocytes by flow cytometry, the cell surface of freshly prepared epidermal cells was labeled with the following anti-human protein anti-bodies (dilution, clone, catalog number, manufacturer): FITC–conjugated CD1a (1:100, HI149, 555806) and PE-conjugated HLA-DR (1:100, G46-6, 555812, both BD Pharmingen), PE-CF594-conjugated CD3 (1:200,UCHT1, 562280) and APC-Cy7-conjugated CD45 (1:200, 2D1, 348815, both BD Biosciences), APC-conjugated CD94 (1:100, DX22, 305508, BioLegend), PE-Cy5.5-conjugated CD117 (1:100, 104D2D1, A66333, Beckman Coulter).

In order to sort CD4$^+$ or CD8$^+$ T-cells, dendritic cell subsets, macrophages, mast cells and endothelial cells from freshly prepared dermal cell suspension the following anti-human antibodies were used: APC-Cy7-conjugated CD45, PE-CF594-conjugated CD3, PE-Cy5.5-conjugated CD117, PE-conjugated HLA-DR, FITC-conjugated CD14 (1:100, MφP-9, 345784, BD Biosciences), FITC-conjugated CD94 (1:100, HP-3D9, 555888, BD Pharmingen), BV421-conjugated FcεRIa (1:100, AER-37, 334624, BioLegend), PE/Dazzle594-conjugated CD4 (1:200, OKT4, 317448, BioLegend), PE-Cy7-conjugated CD8 (1:100, RPA-T8, 557746, BD Pharmingen), PE-conjugated CD11b (1:100, Bear1, PN IM2581, Immunotech), APC-conjugated CD11c (1:100, S-HCL-3, 333144, BD Biosciences), BV605-conjugated CD3 (1:100, OKT3, 317322, BioLegend), BV421-conjugated CD1a (1:100, HI149, 300128, BioLegend), PE/Dazzle594-conjugated CD14 (1:200, HCD14, 325634, BioLegend), PE-Cy7-conjugated CD3 (1:100, SK7, 344816, BioLegend), FITC-conjugated CD31 (1:100, WM59, 303104, BioLegend), PE-Cy7-conjugated CD34 (1:100, 581, 343516, BioLegend). Cells were sorted by FACSAria cell-sorting equipment (BD Biosciences), using an 85-μm nozzle (Supplementary Fig. 3). Sorted cells were washed 2 times with PBS, and finally, cell pellets were immediately snap-frozen in liquid nitrogen and stored at −80 °C until further processing.

**Isolation of PBMCs.** PBMCs were derived from healthy individuals (see "Ethics oversight" above) and separated by density gradient centrifugation over Lym-phoprep (Nycomed-Pharmacia, cat. no. 1114547, Oslo, Norway), collected at the interface and washed twice in PBS.

**Sample preparation for mass spectrometry.** Samples of skin layers were sus-pended in 200 μl, 50 mM Tris-HCl pH8 and individually mixed using an ultra-Turrax blender (T 10 basic ultra, IKA, Staufen, Germany) for 30 s x 8 while kept on ice. 200 μL lysis buffer 2,2,2-trifluoroethanol (TFE) (final concentration 50% [v/v] as well as 5 mM DTT) was added, and the samples were heated for 10 min in the thermomixer. The protocol was slightly modified for the cultivated cells and FACS-sorted cells where there was no need for tissue homogenization. Lysis buffer (containing 20% TFE/ 39,5 mM TRIS HCL/ 5 mM DTT) was added directly to the cell pellets and vortexed for 30 s, and the samples were heated for 10 min in the thermomixer.

The samples were placed shortly on ice and subsequently sonicated using a bioruptor 15 cycles of 30 s. To prevent the disulfide-bonds to restore, IAA were added to a final concentration of 25 mM. The samples were left in dark for 20 min and then dried using vaccum concentrator (#5424 R, Eppendorf, VWR, Søborg, DK) for 30 min at 30˚C. 200 μL digestion buffer (50 mM Tris-HCl, pH 8) was added including Lys-C and trypsin (dependent on the cell number or tissue amount to an estimated enzyme to protein ratio of 1:50), and left overnight in thermomixer for digestion (37 °C, 1300 rpm) to be digested. The following day peptides were acidified 1% trifluoroacetic acid (TFA) to stop digestion and allow for binding to SDB-RPS StageTips. The samples were loaded on 4 layers SDS-RPS material Stage-Tip plugs, and centrifuged at 500 g for 10 min. The samples were washed twice with 0,2% TFA (200 μL), and lastly eluted with 60 μL elution buffer (80% acetonitrile/1% ammonia [v/v]) into a MS-plate (#AB-1300, Thermo Scientific, Life Technologies Europe, Roskilde, DK) and dried using a vaccum concentrator (45 min at 45˚C). Peptides were resuspended in 8 μL loading buffer, 2% acetonitrile in 0.1% TFA and peptide concentration was measured using the Nanodrop®. In addition to the sample processing of each individual sample, we

created matching libraries of pooled samples from the same sample type for high-pH reversed-phase fractionation. About 15–25 μg of peptides were further fractionated into 8 or 16 fractions depending on peptide yield. Supplementary Data 18 provides an overview of the number of fractions in each sample. 500 ng of each unfractionated and fractionated samples was used for subsequent LC-MS analysis.

**LC-MS/MS analysis.** Nanoflow LC-MS/MS analysis of tryptic peptides was con-ducted on a quadrupole Orbitrap mass spectrometer (Q Exactive HF-X, Thermo Fisher Scientific, Bremen, Germany)[15] coupled to an EASY nLC 1200 ultra-high-pressure system (Thermo Fisher Scientific) via a nano-electrospray ion source. 500 ng of peptides were loaded on a 50-cm HPLC-column (75 μm inner diameter, New Objective, Woburn, MA, USA; in-house packed using ReproSil-Pur C18-AQ 1.9-μm silica beads; Dr Maisch GmbH, Ammerbuch, Germany). Peptides were sepa-rated using a linear gradient from 2 to 20% B in 55 min and stepped up to 40% in 40 min followed by a 5 min wash at 98% B at 350 nl per minute where solvent A was 0.1% formic acid in water and solvent B was 80% ACN and 0.1% formic acid in water. The total duration of the run was 100 min. Column temperature was kept at 60 °C using an in-house-developed oven.

For spectral library generation, samples were fractionated using a high pH reversed-phase fractionator as previously described and measured in DDA mode[20]. Briefly, the mass spectrometer was operated in "top-15" data-dependent mode, collecting MS spectra in the Orbitrap mass analyzer (60,000 resolution, 300–1650 $m/z$ range) with an automatic gain control (AGC) target of 3E6 and a maximum ion injection time of 25 ms. The most intense ions from the full scan were isolated with an isolation width of 1.4 $m/z$. Following higher-energy collisional dissociation (HCD) with a normalized collision energy (NCE) of 27, MS/MS spectra were collected in the Orbitrap (15,000 resolution) with an AGC target of 1E5 and a maximum ion injection time of 28 ms. Precursor dynamic exclusion was enabled with a duration of 30 s.

For DIA, the acquisition method consisted of one MS1 scan (350 or 300 to 1650 $m/z$, resolution 60,000 or 120,000, maximum injection time 60 ms, AGC target 3E6) and 32 segments at varying isolation windows from 14,4 $m/z$ to 562.8 $m/z$ (resolution 30,000, maximum injection time 54 ms, AGC target 3E6). Stepped normalized collision energy was 25, 27.5 and 30. The default charge state for MS2 was set to 2.

**MS data analysis.** DDA raw files were processed in the MaxQuant environment (1.6.3.4)[57]. The integrated Andromeda search engine was used for peptide and protein identification at an FDR of less than 1%. The human UniProtKB database (October 2017) was used as forward database and the automatically generated reverse database for the decoy search. "Trypsin" was set as the enzyme specificity. Search criteria included carbamidomethylation of cysteine as a fixed modification, oxidation of methionine, acetyl (protein N-terminus) as variable modifications. We required a minimum of 7 amino acids for peptide identification. Proteins that could not be discriminated by unique peptides were assigned to the same protein group. The "match-between-runs" (MBR) option was enabled. MBR was restricted to samples of the same type (skin primary cells, skin tissue layers). Label free quantification was performed using the iBAQ approach[58]. Proteins, which were found as reverse hits or only identified by site-modification, were filtered out. DIA raw files were analyzed using Spectronaut Pulsar X software (Biognosys, version 12.0.20491.17) and default settings for targeted DIA analysis with the "mutated" as decoy method. Data export was filtered by "No Decoy" and "Quantification Data Filtering" for peptide and protein quantifications. Prior to data analysis, protein quantification values were median normalized.

**Statistical analysis.** All statistical and bioinformatic analyses were performed using Perseus (version 1.6.2.3)[59], the R framework (version 3.6.0, https://www.r-project.org/) and CKG's Analytics core (version 1.0b)[60]. Heatmaps were generated using the open source tool Morpheus (https://software.broadinstitute.org/morpheus). Missing values (those displayed as 0 or "NaN" in the MaxQuant output) were imputed using the MinProb approach (random draws from a Gaussian distribution; width = 0.3 and downshift = 1.8).

**Immunohistochemical analysis.** Immunohistochemical studies were performed on formalin fixed and paraffin embedded sections using the antibodies outlined in Supplementary Data 16. Anti-human K1 and K14 were used for double immuno-labeling experiment and staining, both single and double, were performed on the fully automated instrument Omnis (Dako, Glostrup, Denmark). Between all incubation steps, slides were rinsed in wash buffer. Slides were subjected to dewaxing and subsequently to antigen retrieval using either Target Retrieval Solution (TRS), High pH (Agilent/Dako, Glostrup, Denmark Cat. no GV800) at 97 °C (24 min) or TRS Low pH (Agilent/Dako, Cat. no GV805) at 97 °C (20 min) followed by digestion in pepsin solution (ZytoVision GmbH, Bremerhaven, Ger-many, cat.no. ES-0002-50) at 32 °C (12 min). After pre-treatment, slides were incubated with the primary antibodies CD31 clone JC/70A (Ready-to-use), CD163 clone MRQ-26 (1:100), MART1 clone EP43 (1:100), CK10 clone DE-K10 (1:50), CK14 clone SP53 and polyclonal Collagen III (1:500) at 32 °C (30 min). The reactions were visualized using EnVision™ FLEX /HRP/DAB Detection Reagent

(Agilent/Dako, Cat. no. GV800 + GV809/821) following the manufacturer's instructions. Intensity of the reactions were amplified using either EnVision™ FLEX + Mouse (LINKER) or EnVision™ FLEX + Rabbit (LINKER) depending on the host of the primary antibody. For the polyclonal antibody IL1-β (1:50), reactions were detected using Rabbit-anti-Goat/HRP (Agilent/Dako, Cat. no. P0449) diluted 1:100 in wash buffer and incubated at 32 °C (30 min). Visualization was completed using the DAB substrate reagent from above mentioned Kit (Agilent/Dako, Cat. no. GV800 + GV809/821). Information about the immunohistochemical stainings obtained with courtesy from the Human Protein Atlas[61] can be found in Supplementary Data 17.

**Double immuno-labeling technique**. Dewaxing and pre-treatment was done as described above. Sequential double immuno-staining was performed using anti-human K10 clone DE-K10 (1:50) in combination with anti-human CD14 clone SP53 (1:200). In the first sequence, slides were incubated with K10 for 30 min (32 °C) and reactions were visualized using EnVision™ FLEX /HRP/DAB Detection Reagent (Agilent/Dako, Cat. no. GV800 + 821) following the recommendation of the manufacturer. After the visualization with DAB, slides were subsequently incubated with K14 for 30 min (32 °C). The reactions for K14 were detected with EnVision™ FLEX /HRP Detection Reagent (Agilent/Dako, Cat. no. GV800 + 821) and visualized with Magenta Substrate chromogen system (Agilent/Dako, Cat. no. GV925) following instructions given by the vendor. All slides, both single and double stained, were counterstained with Haematoxylin and mounted with Pertex.

**Reporting summary**. Further information on research design is available in the Nature Research Reporting Summary linked to this article.

## Data availability

The proteomic datasets generated in the current study have been deposited to the ProteomeXchange Consortium via the PRIDE partner repository with the dataset identifier "PXD019909".[62] Proteomics data included in this manuscript can be queried at https://skin.science/. Source data are provided with this paper.

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

## Acknowledgements

The authors thank the skilled technical assistance of Rebeca Soria Romero (Clinical Proteomics, NNF CPR, University of Copenhagen), Jessica Teague (Department of Dermatology, BWH/HMS), and Jeppe Madsen (NNF CPR Mass Spectrometry Platform, University of Copenhagen). Martin Dyring-Andersen helped with the webpage. Sylvie Ricard-Blum (University of Lyon) and Lise-Mette Rahbek Gjerdrum (Department of Surgical Pathology, Zealand University Hospital) shared their knowledge of collagen biology and skin pathology. Bjørn Crewe (Department of Plastic and Breast Surgery, Zealand University Hospital) assisted with sample collection. We thank Juliet Percival for help with illustrations in Fig. 1a–c. This work was supported by grants from the Novo Nordisk Foundation (grant agreement NNF14CC0001 and NNF15CC0001), Lundbeck Foundation (R182-2014-3641), The Max-Planck Society for the Advancement of Science, Aage Bangs Foundation, Leo Foundation and The A.P. Møller Foundation for the Advancement of Medical Sciences and the European Union's Horizon 2020 research and innovation program (Marie Skłodowska-Curie grant agreement No. 846795; F.C.), NIH/NIAMS P30AR069625 (R.C.), and NIH/NIAID R01AI127654 (R.C.).

## Author contributions

B.D.-A., developed the concept, performed sample collection, cell culture experiments, FACS analysis, acquired and interpreted the proteomics data, and wrote the manuscript. M.B.L. performed sample collection, cell culture experiments, FACS analysis, developed the concept, and edited the manuscript. F.C. helped with the study design, acquired and interpreted the proteomics data, and edited the manuscript. A.S.D. helped with the study design, developed and implemented the bioinformatics methods, and edited the manuscript. M.B.M.T. performed sample collection, FACS sorting, and edited the manuscript. L.B.P.M. helped with the sample preparation, acquired the proteomics data. M.B. performed immunohistochemistry. J.L.A. performed sample collection. A.F.C. helped with bioinformatic analyses. L.N. helped with the MS measurements. S.D. helped with the method development. R.A.C. helped with the study design and edited the manuscript. L.S. helped with the study design, interpretation of proteomic data, and edited the manuscript. M.M. designed and supervised the study and edited the manuscript. All the authors have approved the final version

## Funding

## Competing interests

The authors declare no competing interests.
