## [Peer Review File · Nature Communications]

REVIEWER COMMENTS

Reviewer #1 (Remarks to the Author):

The paper written by Dyring-Andersen and colleagues introduces a detailed protein atlas of human skin and skin-resident cell types. Protein abundances of tissues, isolated and sorted cells were analyzed by mass spectrometry (MS)-based proteomics. With more than 10`000 quantified proteins this is the most comprehensive analysis up-to-date. The authors do not present mechanistic data of characterized cell- or tissue-specific proteins. However, they generated a searchable database, which allows interested readers to query their proteins of interest. Thus, the presented data are a great resource and of high value for basic and applied researches working on skin pathophysiology.

The paper is well written and the linked database appears user-friendly. Prior publication, few minor points should still be addressed.

(1) The authors cite several papers that characterized skin-resident immune cells; however, also keratinocytes and fibroblasts are analyzed by MS-based approaches since decades (e.g. see recent review PMID: 32552150 for respective references). Some of the early work should receive credit to underline the long-lasting endeavor to characterize skin and skin-resident cells by MS-based proteomics.

(2) The skin is characterized by a highly complex extracellular matrix (ECM), which is mainly produced by keratinocytes and fibroblasts. The skin ECM played a crucial role for the definition of the "matrisome" (see PMID: 31462529, 22159717).

For readers it would be extremely interesting to know which matrisome proteins are produced by which cell type. Also, matrisome proteins that are only detected in tissue and not in isolated cells (such as "late cornified envelop proteins such as LCE1E, LCE1F and LCE6A and loricrin." mentioned in the manuscript) indicating either differences due to cell differentiation or due to paracrine cell signaling effects would allow interesting insights into skin biology. It would be great if the authors could include respective bioinformatic analyses.

Reviewer #2 (Remarks to the Author):

This is an intense body of work with an incredible amount of data. The authors have developed an innovative and analytic proteomic method that circumvents the presence of highly abundant proteins, such as collagens in the dermis, that make it difficult to detect low abundance proteins and

allows for sensitive measurement of proteins in many cell types that are present in low numbers. The results from this paper are paradigm shifting and will be used by others for years to come.

The authors used a combination of curettage, skin dissection, FACS sorting and primary cell culture to separate the skin into four layers and nine cell types. From these, they generated a proteomic atlas of 10,701 proteins in normal healthy skin.

This is an incredible piece of science, that captures in the text, just a glimpse of what all of the data shows. The Tables are a treasure trove of information that will be mined by others for years.

The methodology is well explained, and the figures are well put together and help the reader visually see what has been done. For example, Figure 4 presents the data in a unique and highly comprehensible way.

The challenge with a body of work such as this, is to determine what to present. The authors have done a great job, but it poses the question, what have they not presented? What is novel? Beyond the approach – have they identified any novel proteins? The clear next step will be to use this technology in skin disease – and perhaps this paper is just the “introduction” to the next one.

The bulk of the innovation lies in their tables and the list of differentially expressed proteins and the cellular subsets they are found within. This is an incredible resource for anyone wanting to know comprehensive protein mapping in the skin and will be a critical resource for years and will be highly cited.

Suggestions and comments:

Regarding the immune cells in the skin – what about B cells?

Include the nerves when talking about skin anatomy – and recognize that their cell bodies lie outside of the skin.

Thanks for the opportunity to review this work,

Nicole L. Ward, PhD.

Reviewer #3 (Remarks to the Author):

In the article entitled “Spatially and cell-type resolved quantitative proteomic atlas of healthy human skin”, the authors have used, among others, data-dependent acquisition (DDA) and data-independent acquisition (DIA) mass spectrometry approaches to generate a comprehensive proteomic atlas of 10,701 proteins in healthy human skin and to describe the quantitative distribution of proteins including those of very low abundant ones, across the skin layers and cell types. These findings are crucial for a better understanding of the functions and roles of skin layers and cells in normal skin function. These proteomic measurements will also be a unique and rich source of information for assessing changes in the proteome content of skin lesions.

The manuscript is recommended for publication pending addressing of the following comments:

Comments:

#1. Page 4: The authors pooled all samples across donors and separated them into eight or 16 fractions by high pH reversed-phase fractionation. The authors may make it clear in the manuscript why and which samples were fractionated into 8 or 16 fractions. This is because different levels of fractionation will result in a different number of protein IDs when processed by mass spectrometry. This will eventually affect the proteomic comparison across the samples studied.

#2. Page 5 and Fig 1e: The authors state that “skin layers and skin-derived immune cells segregated by the first and second component with 24.0% and 11.0% of the total data variation, respectively.” However, looking at the Fig 1e, it appears that not only the immune-derived cells but other cell types including keratinocytes are separated from the skin layer proteome.

#3. Page 7: The authors have used quantitative mass spectrometry data (relative abundance analysis) to describe the spatial gradient of the skin layers’ proteome. However, the link between the datasets used (which I believe is the DDA dataset), the nature of differential abundance analysis performed (i.e. in Fig 2b), and the results outlined here are unclear. The authors should make a clear link between them either in the method or briefly in the result section. It will greatly improve the readability of the manuscript.

Also, it is not clear why the authors did not use DIA analysis for protein quantification in the skin layers (as used for cell types analysis) and then using it to evaluate their relative abundance in different skin layers (i.e. in Fig 2d). If I am correct, the comparative protein expression (or abundance) analysis between the skin layers was based on the abundance of proteins identified and quantified by DDA analysis of skin samples pooled across the donors (page 5).

The X-axis in Fig. 2d is also not labelled.

#4. Page 26: Suppl. Fig. 1. For a better visual comparison between each step in the isolation of skin layers, the authors may use images from the same tissue section and with the same magnification.

#5. Page 35: To better follow the steps taken to process the samples with mass spectrometry, the authors may consider moving the DIA data analysis paragraph after the DDA data analysis.

#6. In the Supplementary Table 2, Gene ID alone has been used as an identifier while in the Supplementary Table 3, UNIPROT ID and GENE ID have been used as identifiers. This is while in the Supplementary Table 6. only UNIPROT ID has been used. For consistency, the authors may use GENE ID as the main identifier and UNIPROT ID as an additional identifier when needed.

#7. Supplementary Table 5. Heading for Column X seems to be incorrect.

#8. The spectral library generated in this study is a rich source of information that can be used for future proteomic studies of skin conditions using DIA mass spectrometry approach. Therefore, the authors may consider making their spectral library publicly available for use by other skin researchers.

Point by Point reply to Reviewer's Comments for "Spatially and cell-type resolved quantitative proteomic atlas of healthy human skin"

Reviewer #1 (Remarks to the Author):

The paper written by Dyring-Andersen and colleagues introduces a detailed protein atlas of human skin and skin-resident cell types. Protein abundances of tissues, isolated and sorted cells were analyzed by mass spectrometry (MS)-based proteomics. With more than 10`000 quantified proteins this is the most comprehensive analysis up-to-date. The authors do not present mechanistic data of characterized cell- or tissue-specific proteins. However, they generated a searchable database, which allows interested readers to query their proteins of interest. Thus, the presented data are a great resource and of high value for basic and applied researches working on skin pathophysiology.

The paper is well written and the linked database appears user-friendly. Prior publication, few minor points should still be addressed.

We thank the reviewer for the positive evaluation of our work and the appreciation of the searchable database that we hope will be a valuable resource for other scientists in the field of dermatology and immunology.

(1) The authors cite several papers that characterized skin-resident immune cells; however, also keratinocytes and fibroblasts are analyzed by MS-based approaches since decades (e.g. see recent review PMID: 32552150 for respective references). Some of the early work should receive credit to underline the long-lasting endeavor to characterize skin and skin-resident cells by MS-based proteomics.

We thank the reviewer for this comment and agree that it is important to give credit to the efforts in the field to provide knowledge about the proteome of the skin-associated cells. We have now included the review by Dengjel et al (Skin proteomics – analysis of the extracellular matrix in health and disease, Expert Review of Proteomics, 17:5, 377-391) as a reference for the efforts in the field on proteomics on primary cells from the skin (page 3, line 45). Novel aspects of our manuscript in this respect are that the primary cultures are from the same donors as the layer data and collected from cells in their first passages.

(2) The skin is characterized by a highly complex extracellular matrix (ECM), which is mainly produced by keratinocytes and fibroblasts. The skin ECM played a crucial role for the definition of the "matrisome" (see PMID: 31462529, 22159717).

For readers it would be extremely interesting to know which matrisome proteins are produced by which cell type. Also, matrisome proteins that are only detected in tissue and not in isolated cells (such as "late cornified envelop proteins such as LCE1E, LCE1F and LCE6A and loricrin." mentioned in the manuscript) indicating either differences due to cell differentiation or due to paracrine cell signaling effects would allow interesting insights into skin biology. It would be great if the authors could include respective bioinformatic analyses.

The matrisome is the ECM proteins and associated factors, and as such, the matrisome, is complex and very important. The skinatlas includes a multitude of these proteins and our strategy has been to present important findings and introduce the reader to the possibilities for further exploration that the data presents. We had therefore included important ECM proteins in Table 2 and Figure 2B such as collagens, laminins, elastin, biglycan, lumican, prolargin. While we agree with the reviewer that a detailed review of the ECM based on the skinatlas data would be both interesting and probably result in novel findings, we believe that the required analyses and interpretation are beyond the scope of this manuscript.

Reviewer #2 (Remarks to the Author):

This is an intense body of work with an incredible amount of data. The authors have developed an innovative and analytic proteomic method that circumvents the presence of highly abundant proteins, such as collagens in the dermis, that make it difficult to detect low abundance proteins and allows for sensitive measurement of proteins in many cell types that are present in low numbers. The results from this paper are paradigm shifting and will be used by others for years to come.

The authors used a combination of curettage, skin dissection, FACS sorting and primary cell culture to separate the skin into four layers and nine cell types. From these, they generated a proteomic atlas of 10,701 proteins in normal healthy skin.

This is an incredible piece of science, that captures in the text, just a glimpse of what all of the data shows. The Tables are a treasure trove of information that will be mined by others for years.

The methodology is well explained, and the figures are well put together and help the reader visually see what has been done. For example, Figure 4 presents the data in a unique and highly comprehensible way.

The challenge with a body of work such as this, is to determine what to present. The authors have done a great job, but it poses the question, what have they not presented? What is novel? Beyond the approach – have they identified any novel proteins? The clear next step will be to use this technology in skin disease – and perhaps this paper is just the “introduction” to the next one.

The bulk of the innovation lies in their tables and the list of differentially expressed proteins and the cellular subsets they are found within. This is an incredible resource for anyone wanting to know comprehensive protein mapping in the skin and will be a critical resource for years and will be highly cited.

Suggestions and comments:

Regarding the immune cells in the skin – what about B cells?

Include the nerves when talking about skin anatomy – and recognize that their cell bodies lie outside of the skin.

Thanks for the opportunity to review this work,

Nicole L. Ward, PhD.

Reply: We thank the reviewer for the positive and constructive comments.

We have revised the introduction by including the sensory nerves on p.3 lines 41-42.

We did not include B cells in the skinatlas due to previous IHC experiments where we did not find B cells in healthy skin (stainings below, Bos JD, and Teunissen MB. 2008. *Innate and Adaptive Immunity In Clinical and Basic Immunodermatology* Gaspari AA, and Tyring SK, eds. Springer, London: 17–30).

Our observations at the time were in line with early reports on the immunophenotypes of skin-associated immune cells and data from the proteinatlas (<https://www.proteinatlas.org/ENSG00000012124-CD22/tissue/skin>, Bos JD et al, 1987. The skin immune system (SIS): distribution and immunophenotype of lymphocyte subpopulations in normal human skin. *J. Invest. Dermatol* 88: 569–573).

More recently, Saul et al (IgG subclass switching and clonal expansion in cutaneous melanoma and normal skin. *Sci Rep* 6, 29736 (2016). have reported skin-associated B cells in healthy skin in FACS experiments. In this study the B cells (defined CD45+CD19+CD22+) made up 1% of CD45+ cells. The low numbers would probably not be sufficient for proteomic studies and might in worst case originate from vessels in the dermis.

CD20 staining of healthy and psoriatic skin

Reviewer #3 (Remarks to the Author):

In the article entitled “Spatially and cell-type resolved quantitative proteomic atlas of healthy human skin”, the authors have used, among others, data-dependent acquisition (DDA) and data-independent acquisition (DIA) mass spectrometry approaches to generate a comprehensive proteomic atlas of 10,701 proteins in healthy human skin and to describe the quantitative distribution of proteins including those of very low abundant ones, across the skin layers and cell types. These findings are crucial for a better understanding of the functions and roles of skin layers and cells in normal skin function. These proteomic measurements will also be a unique and rich source of information for assessing changes in the proteome content of skin lesions.

The manuscript is recommended for publication pending addressing of the following comments:

Comments:

#1. Page 4: The authors pooled all samples across donors and separated them into eight or 16 fractions by high pH reversed-phase fractionation. The authors may make it clear in the manuscript why and which samples were fractionated into 8 or 16 fractions. This is because different levels of fractionation will result in a different number of protein IDs when processed by mass spectrometry. This will eventually affect the proteomic comparison across the samples studied.

Reply: We agree and thank the reviewer for this comment. It is important for us that the readers understand the data preparation and analyses. We used high pH reversed-phase fractionation into 8 or 16 fractions if possible given the peptide yield from each sample. We have added this information to the methods section (P 33, line 602) and created a Supplementary table (Suppl. Table 18). We believe that this creates the maximal depth of protein coverage for the protein atlas overall.

#2. Page 5 and Fig 1e: The authors state that “skin layers and skin-derived immune cells segregated by the first and second component with 24.0% and 11.0% of the total data variation, respectively.” However, looking at the Fig 1e, it appears that not only the immune-derived cells but other cell types including keratinocytes are separated from the skin layer proteome.

Reply: We thank the reviewer for pointing this out. We have now re-phrased the sentence (page 5, line 102).

#3. Page 7: The authors have used quantitative mass spectrometry data (relative abundance analysis) to describe the spatial gradient of the skin layers’ proteome. However, the link between the datasets used (which I believe is the DDA dataset), the nature of differential abundance analysis performed (i.e. in Fig 2b), and the results outlined here are unclear. The authors should make a clear link between them either in the method or briefly in the result section. It will greatly improve the readability of the manuscript.

Reply: We thank the reviewer for this important point. We have now specified that the DDA dataset is used for the relative abundance analysis (page 7, line 128).

Figure legend to Figure 2b has been revised and states: *Relative expression levels of important structural and immunological proteins across skin layers and in primary keratinocytes and fibroblasts. Clustering is based on the log₂ expression level of the proteins in skin layers (DDA).*

In supplementary table 2 we have revised the title to: *Relative expression levels (log₂) of important structural and immunological proteins across skin layers and in primary keratinocytes and fibroblasts based on DDA data.*

Also, it is not clear why the authors did not use DIA analysis for protein quantification in the skin layers (as used for cell types analysis) and then using it to evaluate their relative abundance in different skin layers (i.e. in Fig 2d). If I am correct, the comparative protein expression (or abundance) analysis between the skin layers was based on the abundance of proteins identified and quantified by DDA analysis of skin samples pooled across the donors (page 5).

Reply: Our strategy for the proteomic dataset was to get as deep proteomic coverage as possible. At the time the project was conceived, this implied library construction for DIA for all samples. The latest developments in ‘direct DIA’ increasingly enable DIA on samples without any separate library measurements, and we plan to use this strategy in the future. In this work, we chose to fractionate the skin layers and cellular subsets and create deep libraries for all samples where we had sufficient material. Additionally, we wanted to run single-shot DIA on skin-associated immune cells from multiple donors separately due to their functions and immune phenotypes (identifying CD proteins, cytokines, chemokines). Again, in the future, we plan to analyze skin-layers by DIA for comparisons to diseased skin and the data will eventually be available at the web resource.

The X-axis in Fig. 2d is also not labelled.

Reply: We have labelled the X-axis in Fig. 2D.

#4. Page 26: Suppl. Fig. 1. For a better visual comparison between each step in the isolation of skin layers, the authors may use images from the same tissue section and with the same magnification.

Reply: We thank the reviewer for pointing this out. We have added an image with the same magnification (Suppl. Fig 1b) and added scale bars to Suppl. Fig 1a-c.

#5. Page 35: To better follow the steps taken to process the samples with mass spectrometry, the authors may consider moving the DIA data analysis paragraph after the DDA data analysis.

Reply: We agree with the reviewer’s comment and have moved the paragraph on p. 35 as suggested.

#6. In the Supplementary Table 2, Gene ID alone has been used as an identifier while in the Supplementary Table 3, UNIPROT ID and GENE ID have been used as identifiers. This is while in the Supplementary Table 6. only UNIPROT ID has been used. For consistency, the authors may use GENE ID as the main identifier and UNIPROT ID as an additional identifier when needed.

Reply: We thank the reviewer for pointing out the inconsistency of protein nomenclature in the tables. We prefer to use Uniprot IDs and use gene names when necessary. This is because, the gene name is less precise as genes often produce more than one protein and mass spectrometry offers this resolution. We have now added Uniprot ID to Supplementary Table 2. In this table and in general we have used gene names as a supplement to Uniprot IDs to help the reader where we find it important.

#7. Supplementary Table 5. Heading for Column X seems to be incorrect.

Reply: We thank the reviewer for pointing this out. We have now revised the heading.

#8. The spectral library generated in this study is a rich source of information that can be used for future proteomic studies of skin conditions using DIA mass spectrometry approach. Therefore, the authors may consider making their spectral library publicly available for use by other skin researchers.

Reply: The spectral libraries for the data presented in this manuscript will be publicly available in the PRIDE database, where it is already uploaded. The data can also be queried at <http://skin.science>.